# Examining the Association between Evidence-Based Practice and Burnout among Spanish Physical Therapists: A Cross-Sectional Study

**DOI:** 10.3390/jpm11080805

**Published:** 2021-08-18

**Authors:** Óscar Rodríguez-Nogueira, Raquel Leirós-Rodríguez, Arrate Pinto-Carral, Mª José Álvarez-Álvarez, Jaume Morera-Balaguer, Antonio R. Moreno-Poyato

**Affiliations:** 1SALBIS Research Group, Nursing and Physical Therapy Department, Universidad de León, Astorga Ave. 15, 24401 Ponferrada, Spain; orodn@unileon.es (Ó.R.-N.); apinc@unileon.es (A.P.-C.); mjalva@unileon.es (M.J.Á.-Á.); 2Nursing and Physical Therapy Department, Universidad Cardenal Herrera-CEU, Plaça Reis Catòlics 19, 03204 Elche, Spain; jmorera.el@uch.ceu.es; 3Department of Public Health, Mental Health and Maternal and Child Health Nursing, Universitat de Barcelona, 08907 L’Hospitalet de Llobregat, Spain; amorenopoyato@ub.edu

**Keywords:** occupational burnout, evidence-based practice, quality of health care, person-centered care, occupational diseases, health occupations, clinical competence

## Abstract

The aim of this study was to quantify the level of burnout and competence for evidence-based practice among Spanish physical therapists and to determine if there is a relationship between these and other socio-professional factors. A cross-sectional study with 472 Spanish Physiotherapists. An electronic survey was conducted that included the Maslach Burnout Inventory, Evidence-Based Practice Questionnaire and sociodemographic data. The three subscales of the Burnout correlated with attitude and total Evidence-Based Practice Questionnaire. Attitude and practice for evidence-based practice, educational level and experience were the variables that showed the greatest influence on burnout. Burnout and the degree of evidence-based practice were identified as being discretely related. Specifically, it seems that the evidence-based practice could improve the lack of personal accomplishment, meaning that through interventions perceived as more effective and advantageous, a sense of mastery and self-efficacy is experienced.

## 1. Introduction

Burnout was described four decades ago as a syndrome of emotional exhaustion, depersonalization and reduced self-fulfillment in the work environment [1]. The prevalence of this syndrome is high among healthcare professions that are in the front line of patient care, such as physicians and nurses [2]. Although less researched, physical therapists may also suffer from burnout [3], considering that physical therapy is a profession in which patient interaction plays an important role.

The risk factors for suffering from burnout are multifactorial. Those associated with work can be divided into work-related factors, such as low job autonomy, loss of control over the work environment and lack of time due to administrative requirements [4]; personal characteristics, such as being overly self-critical, unhelpful coping strategies, over-commitment, perfectionism and idealism [5]; and organizational factors, such as workload distribution, incongruence between the values of the work institution and those of the employee, low recognition, insufficient interpersonal collaboration and insufficient opportunities for job advancement [6,7].

Burnout is therefore an important psychosocial problem that gives rise to three characteristic symptoms: emotional exhaustion (EE) causing lack of work motivation, depersonalization (DP), emotional distancing and impersonal treatment, and low personal accomplishment (PA), with a tendency towards negative self-evaluation and feelings of lack of competence [8,9].

Evidence-based practice (EBP) has been defined as the combination of personal clinical experience with the latest and most reliable research evidence [10]. The main characteristics of EBP are confidence and judicious use of the best available evidence, clinical expertise and consideration of patients’ needs and preferences [11]. It has been demonstrated that physical therapy interventions based on EBP have a higher probability of success and lower economic cost and reduce the need for medical-surgical interventions [12,13]. For these reasons, it is currently considered the gold standard method of clinical practice [14].

Although direct evidence is scarce, some authors have associated the implementation of EPB with increased job satisfaction, both in nurses and in the nursing profession [15], as well as among physicians [16,17,18]. In addition, EBP-based care in nursing can improve the attitude of professionals, a key variable in professional satisfaction [19].

Given that job satisfaction and attitude towards work can be two protective factors against burnout, and given the scarcity of evidence on the prevalence of burnout among physical therapists, the present study was considered necessary to quantify the level of burnout and EBP competence of Spanish physical therapists and to determine whether there is a relationship between these factors and other socio-professional factors.

## 2. Materials and Methods

### 2.1. Study Design and Sample

This study consisted of a cross-sectional survey of Spanish physical therapists. The inclusion criteria were physical therapists belonging to one of the professional boards of physical therapy in Spain.

According to the Spanish General Council of Physical Therapy Boards, in 2020, there were 59,592 registered physical therapists in Spain [20]. A sample calculation was made to achieve a margin of error of 5% and a confidence level of 97%. To this end, it was determined that a sample of 468 participants was necessary. Finally, 472 physical therapists belonging to all the boards of physical therapy in Spain participated.

The study was conducted according to the guidelines of the Declaration of Helsinki and approved by the Ethics Committee of the University of León (code: 032-2019).

### 2.2. Procedure

An email was sent to the 17 physical therapy boards in Spain requesting their participation in the study in September 2020. The Google Docs platform was used to create the survey, activating the option of one response per user to avoid duplicate responses. The professional boards that agreed to participate were requested to send the survey link to each of their members via email or publicize the study through their social media accounts (Twitter, Facebook, Instagram).

### 2.3. Instruments 

The electronic form included a questionnaire with the physical therapists’ sociodemographic and professional data and measurement instruments. The variables included were gender, years of experience as a physical therapist, highest education and type, type of work contract and shift.

The measurement instruments included the Spanish adaptation of two validated questionnaires: the Maslach Burnout Inventory Survey in its version for the Human Services (MBI-HSS) [21] and the Evidence-Based Practice Questionnaire (EBPQ-19) [21].

The level of burnout was measured using the MBI-HSS developed by Maslach and Jackson [1]. This instrument consists of three subscales, and the scores for each are classified as low, moderate or high [22]:(a)PA at work: tendency to evaluate oneself negatively regarding work skills and to relate professionally with patients. Formed by 8 items, the resulting score is categorized as follows: less than 30 points, high; 31–36 points, moderate; and 37 or more points, low.(b)EE: the emotional and affective domain. Formed by 9 items, the resulting score is categorized as follows: less than 16 points, low; 17–26 points, moderate; 27 or more points, high.(c)DP: feelings and attitudes of cynicism and negative attitudes towards patients. Evaluated using 5 items, the resulting score categorized as follows: less than 8 points, low; 9–13 points, moderate; and 14 or more points, high.

Evidence-based practice was measured using the EBPQ-19 developed by Upton and Upton [23]. This instrument is made up of three dimensions: (a) practice, which includes six items; (b) attitude, with three items; and (c) professional knowledge and skills for EBP, with 10 items. Each item scores from 1–7, with 1 being the least favorable value and 7 the most favorable regarding the application of EBP.

### 2.4. Data Analysis

A descriptive analysis of all the quantitative variables was performed through calculation of the mean values (to determine the central tendency) and the standard deviation (as a measure of dispersion). The categorical variables were expressed as the number and percentage.

The variables showed a normal distribution according to the Kolgomorov–Smirnov test (*p* > 0.05), and there was homogeneity of variances, applying the Levene test. The association between the quantitative variables was evaluated using Pearson’s correlation coefficient. The relationship between quantitative and categorical variables was determined using the Student’s *t*-test.

Finally, multiple linear regression models were used for the MBI-HSSS subscales as dependent variables and the EBPQ-19. and sociodemographic and professional variables of physical therapists were treated as independent variables.

Statistically significant results were established with a *p*-value of *p* < 0.05 in all analyses performed. The STATA v. 12 statistical package (College Station, TX, USA) was used for statistical analysis.

## 3. Results

### 3.1. Descriptive Analysis

Table 1 shows respondents’ personal and professional characteristics. The mean age of respondents was 33.4 ± 9.3 years, and 69.7% (*n* = 323) were women. The mean years of professional experience of physical therapists was 7.7 ± 8 years, and 34.7% had postgraduate studies.

The employment variables revealed that 69.7% were paid employees, 73.9% worked in the private sector, and the most frequent work schedule was a split working day (43.7%).

The MBI-HSS subscales showed that 46.9% of the participants obtained a high EE score, 42.2% of them reported a high DP score and 41.3% obtained a high PA score (Table 2). If the participants with medium and high scores in the three subscales are added, the results show that 74% of participants suffered EE, 88.2% had DP and 84.1% had PA. In terms of sexes, women showed a significantly lower score on the DP scale.

In relation to the EBPQ-19, participants reported scores between 4.5 and 5.7 points in the three subscales, in decreasing order: attitude, practice, and professional knowledge and skills (Figure 1).

As for the gender difference, men scored significantly higher in the three subscales and in the total.

### 3.2. Correlation Analysis

The subscales of the MBI-HSS obtained statistically significant results in the correlation analysis with the total score of the EBPQ-19 and its subscales. Specifically, the EE score was significantly correlated with attitude (*r* = −0.5, *p* < 0.01), professional knowledge and skills (*r* = −0.5; *p* < 0.05), and the total result of the EBPQ-19 (*r* = −0.6; *p* < 0.05). The DP subscale was correlated with attitude (*r* = −0.6; *p* < 0.01) and the total result of the EBPQ-19 (*r* = −0.6; *p* < 0.05). The PA subscale was correlated with practice, attitude, professional knowledge and skills, and the EBPQ-19 total score (0.7 < *r* > 0.9; *p* < 0.01).

### 3.3. Multiple Linear Regression Analysis

The models generated (Table 3) showed that, of the variables analyzed, those with the greatest influence on EE were the level of studies (with a positive effect) and years of experience and attitude (with a negative effect). DP was conditioned by gender (less impact in women), years of experience and attitude (with a negative effect) and level of studies attained (with a positive effect). Finally, PA was conditioned by years of experience, practice and attitude (with a positive effect), and level of education (with a negative effect).

## 4. Discussion

This study aimed to quantify the level of burnout and EBP competence in Spanish physical therapists and to determine whether there is a relationship between these factors and other socio-professional factors. The analysis of the results showed that the level of burnout of Spanish physical therapists is medium–high, and this phenomenon seems to be related to the degree of EBP and other social and professional factors such as gender, years of experience and level of education. 

The sociodemographic characteristics of the physical therapists in this study are similar to those of other studies assessing the level of burnout in physical therapists in other countries [3,24,25,26,27,28,29], with a majority being women and with a mean age between 30 and 40 years. However, higher levels of burnout have been identified in this study than in previous studies conducted in Arabia [28], United States [3,26], Spain [30] and Cyprus [24].

The findings show that Spanish physical therapists with more years of experience and without postgraduate studies suffer less burnout. Recently, Prentice et al. [31] identified high levels of burnout in postgraduate health care students. In addition, studies involving Spanish physical therapists indicate that they perceive a lack of certain skills and competencies during their undergraduate training [32,33], and a high need for postgraduate training [33], these factors may contribute to the development of burnout. Likewise, high professional qualifications may make it impossible to apply the knowledge obtained due to limitations applied to the job due to stagnant protocols and/or overstretched services [24], lack of time [34] or lack of organizational support [35], which could lead to boredom [34]. This would result in a misalignment with the company, one of the main sources of burnout [36]. Regarding years of experience, Maslach et al. [37] have indicated that the risk of burnout tends to be higher at the beginning of the professional career, due to uncertainty, the perception of a very high workload and a sense of loss of control over the work environment [27].

As in other previous studies [24,26,27], the gender difference in the degree of burnout does not seem relevant; however, the degree of DP was identified as being higher in men. This dimension, closely related to empathy, refers to an attempt to set a distance between oneself and the recipients of the service, making their demands more manageable [37,38]. Several studies suggest that women in health care are more empathetic [39,40,41], which could justify a greater resistance to emotional distancing and impersonal treatment.

The degree of EBP of the physical therapists is consistent with findings in Spanish nurses, who reported the following barriers to daily implementation of EBP: limited time to read research, insufficient authority to change patient care procedures and insufficient time on the job to implement new ideas [42]. These aspects are also applicable to the reality of physical therapists’ work contexts. Regarding the gender differences identified, men obtained higher scores in all subscales and in the total of the EBPQ-19, unlike previous research that reported similar results in the level of knowledge and use of the EBP in men and women [43]. This phenomenon could be due to the fact that the men and women analyzed in this study had a different perception of their capabilities, something that could be common due to gender educational factors [44,45]; this would modify EBPQ-19 scores, given that perceiving that one has a high ability to perform EBP correlates with higher EBP use [43].

Burnout and the degree of EBP were identified as being discretely related. Specifically, it seems that the practice of EBP could improve the lack of PA, meaning that through interventions perceived as more effective and advantageous, a sense of mastery and self-efficacy is experienced [46]. Concurrently, by further implementing EBP-based actions and care, the professional reduces the feeling of uncertainty associated with the health treatments applied [47]. In the same vein, several studies indicate that physicians with higher levels of stress due to uncertainty [48], and those who are reluctant to demonstrate this in front of their patients, have an increased risk of burnout [49].

The influence of EBP on burnout has been identified in other professional groups [46,50,51], yet, to our knowledge this is the first study of this kind in physical therapists. In our sample, we found that an improved attitude towards EBP was associated with better results in the three dimensions related to burnout. This phenomenon could be explained by the fact that positive work attitudes counteract burnout and an improved attitude towards EBP implies a greater willingness to make changes in daily practice [52].

This study has several limitations that should be considered: the cross-sectional design does not allow us to study the evolution of physical therapists’ perceptions over time nor the establishment of causal relationships. Neither the clinical speciality of the physiotherapists surveyed (orthopedic, pediatric, sports, cardiac, neurologic rehabilitation, etc.) nor the number of patients treated per day (or per hour) were taken into account, so these factors should be taken into account as an extraneous variable not analyzed. Future research should also take into account the differentiated analysis of physiotherapists with postgraduate and doctoral degrees. However, this is the first study to determine the prevalence of burnout in Spanish physical therapists and to analyze the relationship between burnout and EBP in physical therapists. Furthermore, the main strength of the findings is the high participation rate and the statistical representativeness of the sample. However, it is likely that the results cannot be extrapolated to other countries in which training plans and care settings differ from those in Spain. 

These results suggest the need to study EBP in-depth as one of the factors influencing burnout, in addition to being a factor that contributes to the quality of the service offered to society. Furthermore, it would be advisable to replicate the study at an international level to confirm the results, develop strategies to improve burnout through EBP and, consequently, improve the quality of care in physical therapy.

## 5. Conclusions

The prevalence of burnout among Spanish physical therapists is 74% of participants with medium of high EE, 88.2% with with medium of high DP and 84.1% with moderate or low perceived PA. Female physical therapists working in Spain are significantly less depersonalized than men, while men have significantly higher scores on all subscales and the total EBPQ-19. Burnout among Spanish physical therapists is directly associated with having limited years of professional experience and postgraduate studies and inversely with EBP attitude and practice variables. 

These findings urge the need to take the necessary decisions to reverse and prevent these adverse outcomes based on multidisciplinary measures from a clinical, managerial and educational point of view. Physical therapists and managers should apply measures to improve satisfaction with the daily clinical practice of physical therapists and, in this manner, improve the level of burnout and the quality of care that society receives from these health professionals.

## Figures and Tables

**Figure 1 jpm-11-00805-f001:**
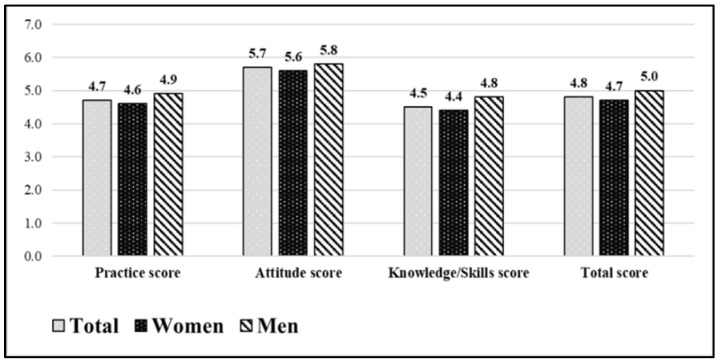
EBPQ-19 scores and levels.

**Table 1 jpm-11-00805-t001:** Participants’ sociodemographic and professional characteristics [data provided: *n* (percentage)].

Variable	All (*n* = 472)	Women (*n* = 323)	Men (*n* = 149)
Age (years):			
20–29	187 (39.6)	115 (61.5)	72 (38.5)
30–39	179 (37.9)	143 (79.9)	36 (20.1)
40–49	88 (18,6)	52 (59.1)	36 (40.9)
50–59	15 (3.2)	12 (80)	3 (20)
60–69	3 (0.6)	1 (33.3)	2 (66.7)
Professional experience (years):			
0–5	165 (35)	98 (59.4)	67 (40.6)
6–10	93 (19.7)	71 (76.3)	22 (23.7)
11–15	81 (17.2)	57 (70.4)	24 (29.6)
16–20	81 (17.2)	60 (74.1)	21 (25.9)
21–25	33 (7)	22 (66.7)	11 (33.3)
26–30	14 (3.0)	12 (85.7)	2 (14.3)
<30	5 (1.1)	3 (60)	2 (40)
Highest education:			
Bachelor’s degree	308 (65.3)	214 (69.5)	94 (30.5)
Ph.D. or master’s degree	164 (34.7)	109 (66.5)	55 (33.5)
Type of work contract			
Self-employed	143 (30.3)	91 (63.6)	52 (36.4)
Employee	329 (69.7)	232 (70.5)	97 (29.5)
Field of work:			
Private	349 (73.9)	231 (66.2)	118 (33.8)
Public	123 (34.7)	92 (74.8)	31 (25.2)
Work shift:			
Morning and afternoon split shift	207 (43.7)	137 (66)	70 (34)
Afternoon shift	99 (21)	61 (61.6)	38 (38.4)
Morning shift	166 (35.3)	125 (75.3)	41 (24.7)

**Table 2 jpm-11-00805-t002:** Scores and levels of the Maslach Burnout Inventory—Human Services Survey.

Variable	All (*n* = 472)	Women (*n* = 323)	Men (*n* = 149)
EE score (mean ± standard deviation)	26.9 ± 10.3	27.2 ± 10.4	26.3 ± 10.2
EE low [*n* (%)]	123 (26.1)	80 (65)	43 (35)
EE medium [*n* (%)]	128 (27.1)	88 (68.8)	40 (31.3)
EE high [*n* (%)]	221 (46.9)	155 (70.1)	66 (29.9)
DP score (mean ± standard deviation)	9.6 ± 4.1	9.3 ± 3.8 *	10.4 ± 4.8 *
DP low [*n* (%)]	56 (11.9)	45 (80.4)	11 (19.6)
DP medium [*n* (%)]	217 (46.0)	143 (65.9)	74 (34.1)
DP high [*n* (%)]	199 (42.2)	135 (67.8)	64 (32.2)
PA score (mean ± standard deviation)	35.2 ± 5.1	38.1 ± 5.1	38.6 ± 5.1
PA low [*n* (%)]	74 (15.7)	52 (70.3)	22 (29.7)
PA medium [*n* (%)]	202 (42.8)	145 (71.8)	57 (28.2)
PA high [*n* (%)]	195 (41.3)	125 (64.1)	70 (35.9)

EE: emotional exhaustion; DP: depersonalization; PA: personal accomplishment; SD: standard deviation; *t*-test results between sexes: * *p* < 0.05.

**Table 3 jpm-11-00805-t003:** Multiple linear regression models for the Maslach Burnout Inventory—Human Services Survey subscales (continuous variables).

Variables Included	Emotional Exhaustion		Depersonalization		Personal Accomplishment	
	B	95% CI	B	95% CI	B	95% CI
Gender 0 = male 1 = female	1.09	−0.89–3.07	−1.09 **	−1.87–−0.29	0.01	−0.88–0.91
Years of experience	−0.16 *	−0.29–−0.02	−0.06 *	−0.12–−0.01	0.17 **	0.11–0.23
Highest education 0 = bachelor’s degree 1 = Ph.D. or master’s degree	2.85 **	0.91–4.78	0.88 *	0.11–1.65	−0.22	−1.09–0.65
Professional knowledge & skills	−0.09	−0.19–0.03	−0.01	−0.06–0.03	0.02	−0.03–0.07
Practice	0.04	−0.13–0.21	0.01	−0.06–0.07	0.19 **	0.12–0.27
Attitude	−0.51 **	−0.87–−0.14	−0.34 *	−0.48–−0.19	0.36 **	0.19–0.53
R^2^	0.072		0.077		0.211	

B: regression coefficient; 95% CI: 95% confidence interval; R^2^: coefficient of determination * *p* ˂ 0.05; ** *p* ˂ 0.01.

## Data Availability

The data presented in this study are available on request from the corresponding author.

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
