# Peer review of "Examining the Association between Evidence-Based Practice and Burnout among Spanish Physical Therapists: A Cross-Sectional Study"

_jpm, 2021, doi:10.3390/jpm11080805_

Round 1
Reviewer 1 Report
The manuscript is well-written and methodologically sound. The authors assessed the prevalence of burnout among physical therapists.
Comments:
- have the authors collected data about the domain the physical therapists worked? Orthopaedic/ paediatric/ sports/ cardiac rehabilitation? It is interesting to find out what domain is related with the burnout.
- what about working hours/ day or the number of patient treated daily? Is there a relation with the burnout?
Author Response
Dear Editor and Reviewer of Journal of Personalized Medicine:
Thank you very much for your suggestions and contributions to improve the quality of the manuscript. Following your indications, we respond, point by point, to the reviewers' comments.
In the text, all the modified or added sentences have been written in red to facilitate the correction by the reviewers.
- Have the authors collected data about the domain the physical therapists worked? Orthopaedic/ paediatric/ sports/ cardiac rehabilitation? It is interesting to find out what domain is related with the burnout.
No, clinical speciality was not included as a study variable in this study. The authors have included it as a limitation of this research.
- What about working hours/ day or the number of patient treated daily? Is there a relation with the burnout?
The aforementioned variables have also not been analysed in this study and, given their importance, will be taken into account in future research and have been included as limitations of this study.
Once again, thank you very much for the time spent and the interest shown in this work; as well as in the positive evaluations you have given of it.
Receive a warm greeting,
The authors.
Reviewer 2 Report
I appreciate the opportunity to review this interesting paper on the association between Evidence-Based Practice (EBP) competence and burnout among Spanish physical therapists.
The genesis of this paper is the hypothesis that EBP competence could affect burnout among Spanish physical therapists. The authors made such a hypothesis based on reports from the literature that implementation of EBM increases job satisfaction which is a protective factor against burnout,
The authors have tested this hypothesis in a cross-sectional study with 472 Spanish physiotherapists. The electronic survey used Maslach Burnout Inventory, Evidence-Based Practice Questionnaire, and questionnaire with the physical therapists’ sociodemographic and professional data.
They found that burnout is directly associated with having limited 316 years of professional experience and postgraduate studies and inversely with EBP attitude and practice variables.
The main strengths of this paper are that it addresses an interesting and timely question, the article is well constructed, the study is well conducted, and the analysis was well performed.
The validated questionnaires used by the authors were appropriate to answer the research question.
Considering these strengths, though, as I read the manuscript, I found some areas in
which I would have appreciated greater clarity. I believe the paper could be further strengthened by added information about:
- Was the tested sample representative of the studied population?
- Can the authors estimate what the response rate has been?
- Have the authors conducted a pilot study?
- The authors analyzed PhD and master degree physiotherapists together in one group. Perhaps the conditions in Spain are different, but my observations show that most physiotherapists in training seek a master degree, but only a few PhD. Therefore, this group seems to be more motivated and more interested in research. This fact can significantly affect their EBP competence and burnout susceptibility.
Author Response
Dear Editor and Reviewer of Journal of Personalized Medicine:
Thank you very much for your suggestions and contributions to improve the quality of the manuscript. Following your indications, we respond, point by point, to the reviewers' comments.
In the text, all the modified or added sentences have been written in red to facilitate the correction by the reviewers.
- Was the tested sample representative of the studied population?
The calculation of the sample size that was used to determine the minimum target participation has been included in section “Study design and sample”.
- Can the authors estimate what the response rate has been?
No, the authors do not have access to this information. The Spanish General Council of Physical Therapy Boards publishes the number of registered physiotherapists but did not provide us with the number of physiotherapists to whom they sent the survey. Therefore, we understand that the number of physiotherapists contacted may be much lower than the number of registered physiotherapists because the Spanish General Council of Physical Therapy Boards does not have the email addresses of all physiotherapists or does not have an up to date email address.
- Have the authors conducted a pilot study?
No, because the instruments used were already validated and the previous literature review sufficiently justified the need for this research.
- The authors analyzed PhD and master degree physiotherapists together in one group. Perhaps the conditions in Spain are different, but my observations show that most physiotherapists in training seek a master degree, but only a few PhD. Therefore, this group seems to be more motivated and more interested in research. This fact can significantly affect their EBP competence and burnout susceptibility.
In this research we only aimed to differentiate between university graduates and those with postgraduate training (regardless of whether they had only a Master's degree or also a PhD).
In any case, the authors have included this variable as a limitation of this research.
Once again, thank you very much for the time spent and the interest shown in this work; as well as in the positive evaluations you have given of it.
Receive a warm greeting,
The authors.